 

# Ligand sensing enhances bacterial flagellar motor output via stator recruitment

**Farha Naaz[†], Megha Agrawal[†], Soumyadeep Chakraborty, Mahesh S Tirumkudulu*, KV Venkatesh***

Department of Chemical Engineering, Indian Institute of Technology Bombay, Mumbai, India

**Abstract** It is well known that flagellated bacteria, such as *Escherichia coli*, sense chemicals in their environment by a chemoreceptor and relay the signals via a well-characterized signaling pathway to the flagellar motor. It is widely accepted that the signals change the rotation bias of the motor without influencing the motor speed. Here, we present results to the contrary and show that the bacteria is also capable of modulating motor speed on merely sensing a ligand. Step changes in concentration of non-metabolizable ligand cause temporary recruitment of stator units leading to a momentary increase in motor speeds. For metabolizable ligand, the combined effect of sensing and metabolism leads to higher motor speeds for longer durations. Experiments performed with mutant strains delineate the role of metabolism and sensing in the modulation of motor speed and show how speed changes along with changes in bias can significantly enhance response to changes in its environment.

**\*For correspondence:**
mahesh@che.iitb.ac.in (MST);
venks@iitb.ac.in (KVV)

[†]These authors contributed equally to this work

**Competing interests:** The authors declare that no competing interests exist.

## Introduction

It is well known that bacteria can execute a net directed migration in response to external chemical cues, a phenomenon termed chemotaxis (*Adler, 1966*; *Berg, 1975*). One of the most commonly studied bacteria for understanding chemotaxis is *Escherichia coli* (*Wadhams and Armitage, 2004*). On sensing a ligand, these bacteria transmit the signal from the environment via a well-characterized signaling pathway to reversible flagellar motors that drive extended helical appendages called flagella thereby achieving locomotion (*Adler, 1969*; *Berg, 2004*). The motor is powered by a proton flux, termed proton motive force (PMF), through an electrochemical gradient of protons across the inner cytoplasmic membrane (*Fung and Berg, 1995*; *Gabel and Berg, 2003*). When all flagella rotate in the counterclockwise (CCW) direction (as seen from the flagellar end), the flagella collect into a single bundle leading to a forward motion called a run, whereas when one or more of the flagella rotate in the clockwise (CW) direction, they separate from the bundle leading to an abrupt change in direction, also known as a tumble (*Berg and Brown, 1972*). By modulating the duration of runs and tumbles, the bacteria achieve a net motion toward chemoattractants or away from chemo-repellents (*Brown and Berg, 1974*).

The direction of rotation of the flagellar motor is controlled by the switch complex comprising of proteins FliG, FliM, and FliN, which are present at the cytoplasmic end of the motor (*Figure 1*; *Yamaguchi et al., 1986*). When CheY-P, the phosphorylated state of the chemotaxis protein CheY, binds to the motor protein FliM, the motor is biased in the CW direction (*Welch et al., 1993*). The null state of the motor, however, is the CCW direction. When a ligand binds to a chemoreceptor protein present in the periplasm, the activity of the chemotaxis pathway protein CheA is reduced, which in turn reduces the production of phosphorylated CheY-P. Since fewer CheY-P molecules are available to bind to the motor protein FliM, the flagella rotate in the CCW direction

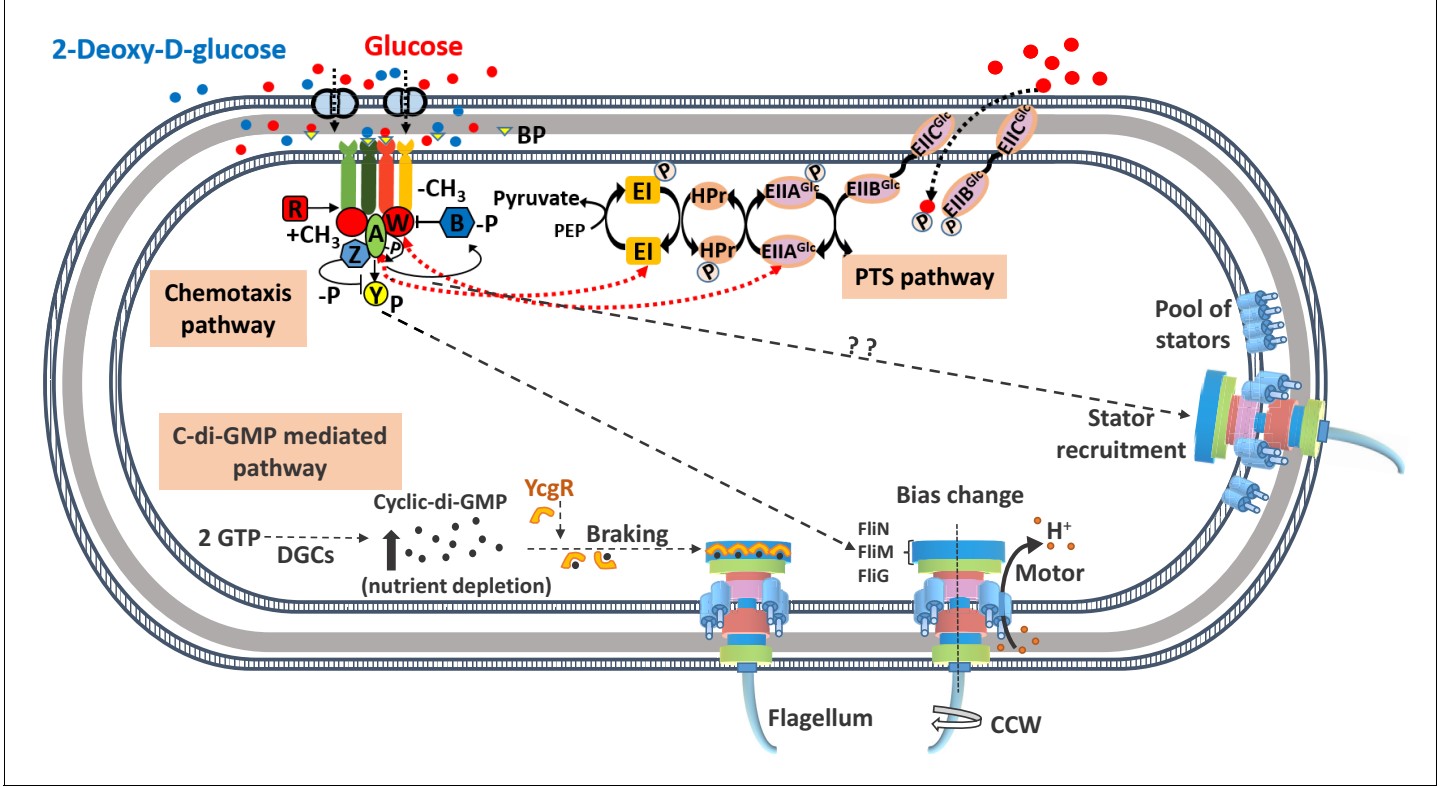

**Figure 1.** A schematic diagram of the known signaling pathways for control of locomotion in *Escherichia coli*. The chemotaxis pathway controls the direction of motor rotation via CheY protein which, on phosphorylation, binds to the FliM protein of motor-switch complex (FliG, FliM, and FliN) and causes clockwise (CW) rotation of one or more flagella. The concentration of phosphorylated CheY (CheY-P) is dictated by the binding of ligands to their specific transmembrane receptors. Attractants such as glucose/2Dg bind to the periplasmic glucose-binding protein (BP) and are sensed by the Trg chemoreceptor protein. The binding results in a decrease in the autophosphorylation activity of CheA, which is part of CheA-CheW-receptor complex, leading to a lower concentration of CheY-P. The CheY-P binds to the FliM protein causing the motor to rotate in the CW direction. The proteins CheR and CheB control the degree of methylation of the receptors and allow the cell to adapt to the present concentration of attractant and sense subsequent changes. Additionally, glucose is a metabolizable ligand and is also sensed by an alternate pathway, namely, the phosphotransferase system (PTS), whose signals integrate into the chemotaxis pathway. The PTS generates a constant flow of phosphate groups via the PTS transporters (EI, HPr, and EII) from phosphoenolpyruvate (PEP) and activates the sensory complexes CheA-CheW thereby communicating with the motor via reduced levels of CheY-P. The CheZ protein accelerates the dephosphorylation of CheY-P. The motor rotation is generated via ion flow through the membrane-bound stator units comprising of proteins, MotA and MotB, while the speed is controlled via the intracellular concentration of the second messenger molecule, cyclic diguanosine monophosphate (c-di-GMP). The latter is produced by diguanylate cyclase (DGC) enzymes from GTP and degraded by specific phosphodiesterases. Nutrient depletion conditions trigger production of c-di-GMP, which forms complex with YcgR protein and hinders the motor rotation. The present study suggests a signaling pathway that links the chemotaxis receptor to the motor speed via stator unit recruitment.

(*Blair, 1995*). For ligand concentrations in the nanomolar to the micromolar range, a feedback control mechanism intrinsic to the signaling network brings back the phosphorylation state of CheA and CheY to the pre-stimulus levels and resets the duration of CW and CCW rotations to the initial state, a characteristic feature of the network termed perfect adaptation (*Masson et al., 2012*).

The torque of the flagellar motor is generated via ion flow through at least 11 membrane-bound stator units comprising of proteins, MotA and MotB (*Yuan and Berg, 2008*; *Reid et al., 2006*; *Khan et al., 1988*). Previous studies have demonstrated the discrete nature of torque generation by restoring the motility of paralyzed cells (*mot* mutants) by induced expression of wild-type (WT) genes from plasmids. The latter caused a step-wise increase in motor speed, a process termed 'resurrection' (*Block and Berg, 1984*; *Blair and Berg, 1988*). Recent experiments have shown that the number of bound stator units vary with PMF with almost all stator units leaving the motor following removal of PMF (*Tipping et al., 2013b*). Consequently, the bacteria is able to vary the number of motor-bound stator units depending on the external mechanical load (*Tipping et al., 2013a*; *Lele et al., 2013*; *Nord et al., 2017*; *Wadhwa et al., 2019*) although not all binding sites are

necessarily occupied even at high load conditions due to the transient nature of stator unit binding. In addition to the torque variation with load, the speed of the motor is also controlled via the intracellular concentration of the second messenger cyclic diguanosine monophosphate (c-di-GMP), whose concentration increases during nutrient depletion conditions (*Boehm et al., 2010*; *Paul et al., 2010*). The c-di-GMP binds to the YcgR protein, a PilZ domain protein, which in turn interacts with the MotA to reduce the rotation speed of the motor. While the aforementioned studies have demonstrated the modulation of motor *bias* in response to sensing of ligand and change in motor speed with PMF or when exposed to starvation conditions, there is as yet no clear evidence of the chemosensory transduction system influencing the motor speed in *E. coli*.

In this study, we investigate the role of sensing and metabolism of glucose on the motor performance by both measuring the motor speed and the concentration of stator units bound to the motor via tethering experiments and confirming the same through run speed measurements in a population. For *E. coli*, glucose is the preferred carbon source for cell's growth and there exist detailed studies on the molecular mechanism of sensing and adaptation to the ligand (*Hazelbauer and Engström, 1980*; *Lux et al., 1999*; *Feng et al., 1999*; *Li and Hazelbauer, 2004*; *Neumann et al., 2012*). The signaling is achieved through two independent pathways: through the phosphotransferase system (PTS) pathway and via the Trg-sensing pathway. The combined signals from the two pathways determines the phosphorylation state of CheA, which in turn determines the concentration of CheY-P, and therefore the motor bias. In order to differentiate the roles of sensing and metabolism on motor performance, we performed identical experiments with glucose and the non-metabolizable analog of glucose, 2Dg. *Adler et al., 1973* have shown that 2Dg is sensed by the Trg sensor, but unlike glucose, are not metabolized since cells do not grow in its presence. Further, cells lacking Trg show chemotactic response to 2Dg only when grown in the presence of glycerol through the activation of mannose PTS pathway (*Adler and Epstein, 1974*).

The experiments were performed on the wild-type strain along with mutants lacking *trg* ($\Delta trg$) and *ptsI* ($\Delta ptsI$) genes. The influence of ligands on metabolism was assessed by measuring the membrane potential which corresponds to PMF in a population of cells. The data from the tethered cell experiments show a step increase in the motor speed when exposed to 2Dg. We also simultaneously measure the intensity of fluorescently tagged stator units bound to the motor (mutant strain JPA804) where increase in intensity indicates increase in number of stator units. Our results indicate that mere sensing of a ligand can directly enhance the motor speed wherein the enhancement occurs due to an increase in the number of stator units bound to the motor. These results clearly demonstrate the existence of a sophisticated signaling pathway that connects the chemotaxis signaling system to the torque-generating stator units.

## Results

### Sensing leads to increase in motor speed

In order to correlate the measured run speed with motor response, tethered cell experiments were performed. *Figure 2A and B* presents the time evolution of the rotation speed of a single motor after the introduction of glucose and 2Dg, respectively. In case of glucose, there is a sharp increase in rotation rate from an average speed of about 6 to 7.5 Hz at about 100 s and again a further increase to about 9 Hz at around 650 s. In comparison, for 2Dg the rotation speed remains unchanged for about 300 s after which there is a sudden increase in average speed from about 4 Hz to about 5.7 Hz. The increased speed is sustained for about 200 s after which it returns to the pre-stimulus levels. The step increases, which range between 1.0 and 2.0 Hz, are of magnitude similar to those observed in the resurrection experiments (*Block and Berg, 1984*; *Blair and Berg, 1988*).

### Sensing leads to stator unit recruitment

The increase in the motor speed may be attributed to the recruitment of stator (MotAB) component of bacterial flagellar motor thereby increasing the torque on the motor. Therefore, GFP-tagged MotB component of stator for tethered mutant cell (JPA804) was directly visualized under a confocal microscope and the intensity variation at the motor site was recorded after the introduction of ligand in the medium (*Figure 2C*). While there is no change in the intensity for a cell in MB, the intensity increases with time when glucose is introduced. This observation is in agreement with previously



**Figure 2.** Effect of sensing and metabolism on stator unit recruitment and motor speed of a single motor. The motor rotation speed was measured after a *single* wild-type (WT, RP437) cell was exposed (at $t = 0$) to 1000 μM of (**A**) glucose and (**B**) 2Dg. The arrows indicate the time of introduction of the ligands. The reported values of rotational rate are averaged over 0.16 s (10 frames). Imaging is performed about 50 s after the introduction of the ligand to let the accompanying flow to stop. (**C**) The fluorescence intensity of the stator units (GFP-MotB in mutant strain JPA804) was measured at different time points after introduction of ligands, and (**D**) presents the corresponding motor rotation speeds (arrow indicates the time of introduction of ligand).

The online version of this article includes the following source data and figure supplement(s) for figure 2:

**Source data 1.** Motor speed data for a single cell.
**Figure supplement 1.** Illustration for motor speed calculation for a tethered cell.
**Figure supplement 2.** GFP-MotB intensity calculation using ImageJ.

reported experiments where the stator unit concentration increases with PMF (*Tipping et al., 2013b*). Interestingly, on introduction of 2Dg, which is a non-metabolizable ligand, the intensity increases for a short duration (about 4–5 min) after which it falls to the pre-stimulus value. This result matches with the change in motor speed (*Figure 2D*) confirming that the modulation in motor speed is indeed caused by recruitment of stator units upon mere sensing of ligand. Similar experiments were performed on over 20 motors and the relative average intensity values at fixed time points are plotted in *Figure 3A*. An increase of 17% at 5 min and 20% increase at 10 min was observed in the fluorescence intensity for glucose relative to its pre-stimulus value. A slightly lower increase (14%) was observed at 5 min for 2Dg after which the intensity decreased to its pre-stimulus value. As expected there was no discernible trend in the intensity value measured for cell exposed to MB. *Figure 3B* presents the corresponding motor speeds for the cells, which show the same trend as



**Figure 3.** Effect of sensing and metabolism on stator unit recruitment, motor speed, membrane potential and swimming speed in a population of cells. (**A**) Fluorescence (GFP-MotB) intensity variation in mutant strain JPA804 in 1000 µM glucose, 1000 µM 2Dg and motility buffer (MB). Measured intensity of individual cells was first obtained in MB and then measured again for the same cells exposed to 1000 µM of glucose and 1000 µM 2Dg (arrow indicates the point of introduction of ligand). Increase in GFP intensity in glucose at all time points and in 2Dg at 5 min is statistically significant at

*Figure 3 continued on next page*

*Figure 3 continued*

p<0.01 computed by paired t-test. The error bars represent standard error of means from six independent experiments. The intensity of about 21–22 motors per ligand were measured. (B) The normalized average motor speed in counterclockwise (CCW) bias obtained from the tethered cell experiments presented in (A). (C) Motor intensity versus motor speed from data plotted in (A) and (B). The straight line is a linear fit with an $R^2$ value of 0.91 (D) Time-dependent fluorescence intensity for membrane potential of wild type (WT) in the presence of 1000 µM of glucose, 2Dg, and sodium MB for a population ($3.75 \times 10^8$ cells/ml). The error bars represent standard error of means from three independent experiments. Values are significant at p<0.001 (one-way ANOVA). (E) Measured run speed between consecutive tumbles for a population of WT cells in the presence of MB and when exposed to ligands. Each data point is obtained by averaging over 2500 cells. Speed increase in glucose for all three strains and the observed increase in 2Dg for the WT strain (at 5 min) as compared to MB are statistically significant at p<0.001 (one-way ANOVA). The Y-error bars represent standard error of means from four independent experiments. (F) Comparison of motor speed in CCW bias for the WT cell in the presence of ligands with those measured for mutant strains, Δtrg and ΔptsI at fixed time points. Data represents value from at least 20 paired cells for each strain. Here, *** represents significance value p<0.001 and ** represents p<0.01 as measured by paired t-test. Error bars represent standard deviation. The highest speed for 2Dg was observed at 5 min.

The online version of this article includes the following source data and figure supplement(s) for figure 3:

**Source data 1.** GFP-MotB intensity, membrane potential, and swimming speed data for wild-type (WT) cells.
**Figure supplement 1.** Response of wild type (WT) after 5 min of exposure to 1000 µM of glucose.
**Figure supplement 2.** Rotation speed normalized with respect to the pre-stimulus value and the counterclockwise (CCW) bias of wild type (WT) at discrete time points in the presence of (A,B) 1000 µM glucose and (C,D) 1000 µM 2Dg.
**Figure supplement 3.** Rotation speed in counterclockwise (CCW) direction for wild-type (WT), Δtrg, and ΔptsI cells.
**Figure supplement 4.** Counterclockwise (CCW) bias for the wild-type (WT), Δtrg, and ΔptsI cells measured in motility buffer (MB), and 1000 µM of glucose and 2Dg.
**Figure supplement 5.** Variance of rotation rate for different number of revolutions determined at different times after exposing a single wild-type (WT) cell to 1000 µM of (A) glucose and (B) 2Dg.
**Figure supplement 6.** RP437 cells exhibiting pH homeostasis.
**Figure supplement 7.** Agarose gel for PCR product verification.

that for the stator intensity. The normalized fluorescent intensity values in different motors for the three conditions and the corresponding normalized motor speeds are presented in *Figure 3C*. The intensity is proportional to motor speeds suggesting that stator elements are added following the introduction of the two ligands.

Following *Samuel and Berg, 1995*; *Samuel and Berg, 1996*, measurements of the variance in rotation speed of the tethered cells were also made as a function of mean number of revolutions to correlate response with number of active torque-generating units. A large variance corresponds to fewer stator units powering the motor while a lower variance and therefore a smoother rotation implies larger number of stator units. The measured variance followed the same trend as observed for the rotation speed and fluorescent stator intensity values in that the lowest variance was found in cells exposed to glucose followed by 2Dg and the highest in MB (*Figure 3—figure supplement 5C–E*). The time evolution of the variance also matched with motor speed and stator intensity data.

## Sensing does not affect membrane potential

As discussed in the Introduction, the stator unit recruitment has been directly correlated with PMF. To confirm that the observed increase in the presence of 2Dg was not due to changes in PMF, we measured the change in transmembrane electrical potential (Δψ) and pH gradient (ΔpH) in the presence of glucose and 2Dg. Since *E. coli* is a neutraphile, the change in the PMF is primarily due to changes in Δψ and not ΔpH (*Figure 3—figure supplement 6*; *Krulwich et al., 2011*). *Figure 3D* presents fluorescence intensity values for WT, which are proportional to membrane potential. The buffer solution in these experiments contains sodium salts (NaMB) instead of potassium since the latter has been shown to interfere with membrane potential measurements (*Kashket and Barker, 1977*; *BacLight, 2004*). The measured potential is the lowest for the control, namely, CCCP (carbonyl cyanide *m*-chlorophenyl hydrazone), which is an ionophore and eliminates the membrane potential. While there is no change in the potential for NaMB and 2Dg and their values are very close, the intensity value in the presence of glucose is significantly higher. This clearly demonstrates that the stator unit recruitment in the presence of 2Dg is independent of PMF and is caused solely by sensing of 2Dg. For glucose, the increase in motor speed would be a combination of increased PMF and that due to sensing.

## Average swimming speed in a population

Given the above observations, does the single motor behavior influence the swimming speed of cells? In order to address this question, run speeds were measured at constant ligand concentration at four different time points, namely, 1.5, 6.5, 11.5, and 16.5 min after the cells were exposed to ligands (average of over 2500 cells). In the presence of glucose, the run speed increased monotonically (*Figure 3E*) with almost a 50% increase (29.6±1.0µm/s, mean ± s.e.) at 16.5 min compared to that measured at the start of the experiment (20.1±0.9µ m/s). In case of 2Dg, the run speed increased upto 6.5 min reaching a value identical to that observed for glucose at the same time, resulting in a 26% increase (25±1.4µm/s) compared to the speeds measured at the start of the experiment. However, at longer times, unlike in glucose, the run speed decreased and reached values observed at the start of the experiment. In the absence of ligands, the run speed showed negligible change with speed values ranging approximately between 18 and 21 µm/s for the entire duration of the experiment. These results indicate that sensing alone can enhance the run speed, albeit for a short duration, as observed in case of 2Dg. A further increase in run speed is observed in case of glucose suggesting that metabolism plays a role in sustaining a high run speed. Clearly, the observations made on a single motor via tethered cell experiments indeed reflect in the population behavior wherein stator unit recruitment due to sensing alone can enhance the swimming speed of a population. We note in passing that the concentrations of the ligands are sufficiently low that osmotic effects do not influence the motion (*Rosko et al., 2017*).

## Role of Trg receptor and PTS in speed modulation

In order to characterize the role of Trg receptor, a mutant strain lacking in *trg* was exposed to both glucose and 2Dg and motor speed was recorded at various times (*Figure 3F*). There was no difference in response between MB and 2Dg for the $\Delta trg$ mutant strain. However, in glucose, there was significant increase in the motor speed (about 30%) compared to the pre-stimulus value. This is in contrast to that observed in WT where the motor speed increased in both glucose (about 40%) and 2Dg (about 30%). These results clearly show that in the absence of the Trg sensor, the motor speed remains unaffected in the presence of 2Dg thereby reconfirming the earlier observation that sensing alone can contribute to motor speed modulation. Experiments measuring the membrane potential show a behavior similar to that observed for WT cells (*Figure 4A*). This is expected since 2Dg is not metabolized and hence does not contribute to PMF and consequently, presence or absence of Trg sensor has no effect on PMF in the presence of 2Dg. The mutant strain exhibits increased PMF in glucose through metabolism although the ligand is not sensed. This behavior is observed at the population level (*Figure 4B*), wherein there is no change in run speed in response to 2Dg, which is identical to that observed in MB, while there is a significant increase in the presence of glucose (about 50% increase).

In order to estimate the role of metabolism in motor speed modulation, we performed the same set of experiments in mutants lacking the *ptsI* gene. There was a marginal increase in motor speed (*Figure 3F*) and run speed (*Figure 4D*) in the presence of glucose while no significant change was observed in the presence of 2Dg. It can be noted that the membrane potential dropped drastically due to the absence of *ptsI* gene (*Figure 4C*) compared to that observed in either WT or $\Delta trg$ strain. The small increase in membrane potential measured for the mutant strain in the presence of glucose explains the marginal increase in the motor and run speed.

## Role of CheY

To determine whether CheY is necessary for the observed increase in run speed (*Figure 3E*), experiments were also performed with a strain lacking CheY ($\Delta cheY$ mutant). These mutant cells do not tumble but always run since the flagellar motors rotate in the CCW direction in the absence of CheY-P. The results of such experiments are shown in *Figure 4F*. The trends are very similar to those observed with WT (*Figure 3E*) in that the run speed increases monotonically with time in the presence of glucose while the speed first increases and then decreases in case of 2Dg. The similarity of the observed trend with WT indicates that CheY-P has no role in the observed modulation in motor speed and the observed speed increases are independent of switching events. The trend in membrane potential for $\Delta cheY$ in the presence of both ligands was similar to that observed in the WT strain (*Figure 4E*).



**Figure 4.** Time-dependent fluorescence intensity for membrane potential and time-dependent run speed variation for mutant strains, (A,B) Δ*trg* RP437, (C,D) Δ*ptsI* RP437, and (E,F) Δ*cheY* RP437 in the presence of MB, and 1000 µM of glucose and 2Dg. Final concentration of cells for membrane potential measurement was $3.75 \times 10^8$ cells/ml and the values for glucose compared to that for MB are significant at p<0.001 (one-way ANOVA). Each data point in the run speed plot is obtained by averaging over 2500 cells. Speed increase in glucose for all three strains and the observed increase in 2Dg for the

*Figure 4 continued on next page*

*Figure 4 continued*
wild-type (WT) strain and CheY mutant (at 5 min) as compared to motility buffer (MB) are statistically significant at p<0.001 (one-way ANOVA). The Y-error bars represent standard error of means from four independent experiments, while the X-error bars represent the duration of 3 min over which the measurements were made for each data point.
The online version of this article includes the following source data for figure 4:

**Source data 1.** Membrane potential and swimming speed data for different mutants.

## Discussion

It is well known that *E. coli* modulates its motor bias in response to ligands (*Adler, 1966*). For example, in case of glucose, which is sensed via the Trg chemoreceptor, sensing leads to the dephosphorylation of CheY-P which in turn reduces the motor reversals leading to smooth runs. Glucose is also sensed and consumed via the PTS pathway, which is known to regulate CheY-P, thereby altering the bias. *Neumann et al., 2012* have shown that the signals from the Trg receptor and the PTS pathway combine (additive) to determine the reversal frequency.

Although it is well known that the chemotactic pathway modulates the motor bias, the existence of a pathway that also modulates the motor speed is not known. Previous studies have shown that in tethered cell experiments, where the torque is in the high load regime, the motor speed is proportional to the ion motive force in both proton and sodium-driven motors (*Gabel and Berg, 2003*; *Sowa et al., 2003*). These changes in motor output have been attributed to the change in the number of stator units, which are assembled from a mobile pool of MotB molecules over a timescale of minutes when the PMF is increased (*Reid et al., 2006*; *Leake et al., 2006*). In contrast, for cases where the mechanical load is low, such as in bead assay with sub-micron sized beads, the motor output can also vary with the magnitude of the load due to load-dependent stator unit recruitment (*Tipping et al., 2013a*; *Lele et al., 2013*). Thus this dynamic engagement of stator units has been shown to modulate motor speed in response to both nutritional status and mechanical load although the exact mechanism responsible for them is not known. While these studies clearly indicate the role of metabolism and mechanical load in motor speed modulation via stator unit recruitment, there is no evidence hitherto that mere sensing of a ligand can directly influence the motor speed at a fixed mechanical load. The tethered cell experiments reported here clearly demonstrate the recruitment of stator units in response to sensing of a ligand and therefore suggest the existence of a pathway that links the sensor to the motor.

Our experiments show an increase in stator unit density at the motor in response to 2Dg, a non-metabolizable analog of glucose. This increase in stator unit density led to a concomitant enhancement in motor speed and thereby the run speed in a population. Since 2Dg is sensed by the Trg sensor, this indicates that a signal for speed modulation is communicated to the motor from the sensor. Further, there was no change in membrane potential confirming the absence of role of metabolism and thereby suggesting that any change in the motor speed is only due to the sensing mechanism. This was further confirmed by deleting the *trg* gene, which eliminated changes in motor speed in the presence of 2Dg. In case of glucose, the enhancement in motor speed is much greater compared to that in 2Dg due to the combined effect of sensing and metabolism, caused by an even greater recruitment of stator units (as indicated by increased intensity of fluorescently tagged stator unit). Here, the increase in motor speed is due to glucose metabolism leading to increased PMF (*Schwarz-Linek et al., 2016*) while that due to sensing is through the Trg receptor. When the metabolism was disrupted (Δ*ptsI* strain), the basal values of the motor speed reduced significantly. A small increase in motor speed was observed when the cell was exposed to glucose but changes were statistically insignificant in the presence of 2Dg. The weak response observed for the Δ*ptsI* strain may be due to the impaired chemotaxis pathway since PTS genes are known to control the expression of proteins involved in chemotaxis (*Neumann et al., 2012*).

An alternate method to characterize stator unit recruitment is to determine the variance in mean rotation rate of the motor as a function of the number of revolutions (*Samuel and Berg, 1995*; *Samuel and Berg, 1996*). Thus, a smaller variance would indicate an increased number of stator units, thereby making the motor rotation smoother. The tethered cell experiment with the WT cells shows clearly a decrease in variance when exposed to 2Dg (*Figure 3—figure supplement 5B and E*) compared to that in MB. The variance decreased even further when exposed to glucose indicating a

higher recruitment of stator units due to the combined effect of metabolism and sensing (*Figure 3—figure supplement 5A and D*). These observations are consistent with the stator unit density measured via fluorescence.

Could the observed modulation of motor speed be controlled by the well-known chemotaxis pathway where the phosphorylated CheY binds to the motor to modulate not only motor bias but also motor speed? To answer this question, motor speed of a mutant lacking in cheY was measured in the presence of both 2Dg and glucose. The observed motor speed modulation was identical to that for the WT indicating that CheY does not play role in stator unit recruitment. These observations clearly suggest that there exists a signaling pathway, independent of the chemotaxis pathway, that links the sensor to the motor. Note that in the cheY mutant, the bias is locked in the CCW direction and therefore is also always in the run mode. Experiments with WT cells on the other hand show a momentary increase in CCW bias after which the bias recovers to the pre-stimulus value thereby demonstrating the well-known chemotactic response. This was observed for both glucose and 2Dg albeit a larger increase in CCW bias in the former (*Figure 3—figure supplement 4*).

The above results may be understood by considering the state of the motor on the torque-speed curve (*Figure 5A*). The dashed line originating from the origin is the load path of the motor under high load conditions. When the cell is exposed to 2Dg, one or more stator units are recruited temporarily (states 1–2 in *Figure 5A*, blue line) thereby increasing the rotation speed (see schematic in *Figure 5B*). The stator units disengage within a short time (about 5–10 min) after which the rotation rate returns to the pre-stimulus values (back to state 1). This effect is due to sensing alone. However, when the cell is exposed to glucose, a metabolite, the combined contribution from sensing and PMF leads to a much larger increase in the number of bound stator units (states 1–3, red line) and the speeds remain high for a much longer time periods (over 15 min), due to the metabolism.

At the population level, these results imply a more efficient chemotaxis response since the drift velocity of a population of cells in response to a ligand gradient is dependent on both the tumble frequency and the swimming speed. Specifically, the well-known RTBL model of *Rivero et al., 1989*, which is based on the Keller-Segel model, links the population-level chemotactic response to the individual microscopic variables of cell's swimming speed, persistence time, and temporal receptor occupation. The drift velocity (also known as the chemotactic velocity) of the population is directly proportional to the product of the swimming speed and a function that depends on the tumbling frequency, spatial gradient of chemoattractant, and occupancy of chemoreceptor (Equation 22 of *Rivero et al., 1989*). For small gradients, the expression simplifies such that the drift velocity varies

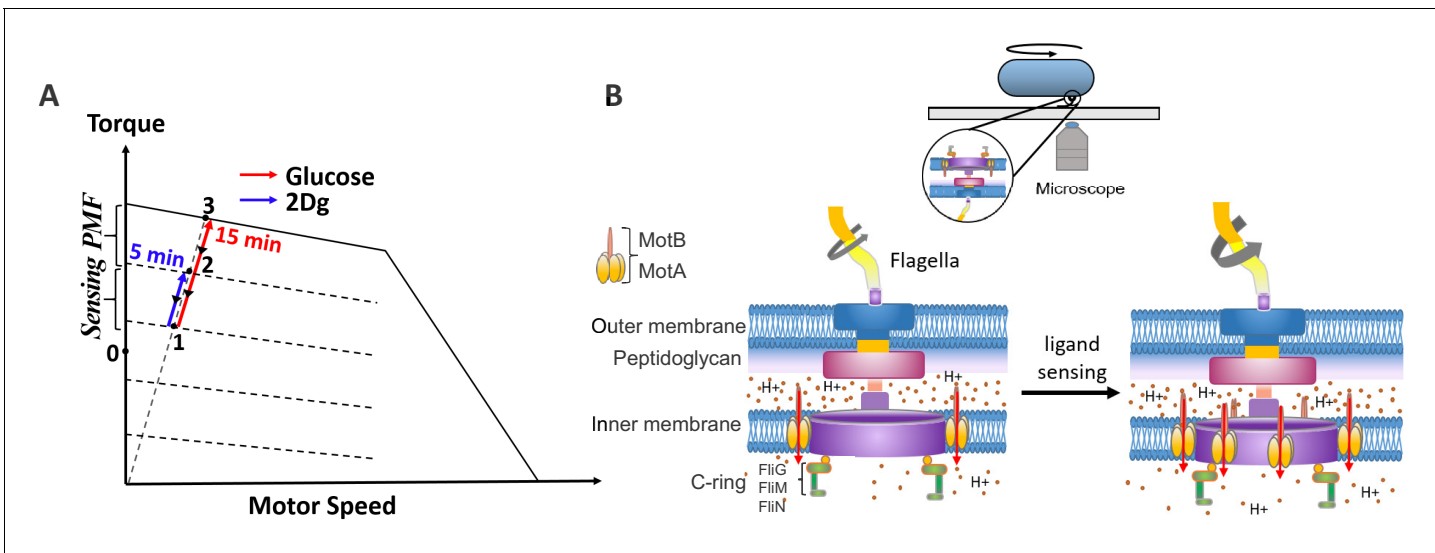

**Figure 5.** Schematic diagram on the effect of sensing and metabolism on the state of the flagellar motor. (**A**) A representative plot of steady-state torque-speed curves for counterclockwise (CCW) rotation of the bacterial flagellar motor at varying loads along with the expected state of the motor along the load line at high load when the cell is exposed to 2Dg and glucose. (**B**) A schematic of the bacterial motor along with a cell tethered to a glass coverslip is included to the right. Motor speed increases as a result of stator unit recruitment upon ligand sensing.

as the square of the swimming speed (Equation 25 of *Rivero et al., 1989*). These results imply that even a 10% increase in the swimming speed could cause the drift velocity to increase anywhere between 10% and 20% leading to a significant improvement in the chemotactic efficiency.

Finally, the pathway is independent of the known sensory transduction pathway since the observed speed modulation also occurs in the absence of CheY (*Figure 4F*). Recent studies have shown that c-di-GMP effector YcgR can inhibit flagellar motility by directly interacting with the motor to modulate both its bias and speed (*Boehm et al., 2010*; *Paul et al., 2010*; *Nieto et al., 2019*). While the aforementioned mechanism has been shown to inhibit motor speeds, there is no evidence that the same can enhance motor speeds beyond pre-stimulus levels. Irrespective of the details of the mechanism, the results presented here point to a sophisticated signaling network with multiple strategies to regulate motor behavior enabling an efficient response to changes in bacteria's environment.

## Materials and methods

### Bacterial strains and growth condition

*E. coli* RP437 and its four mutants (see *Table 1*) were inoculated from glycerol stock into tryptone agar plates. A single colony isolate was grown overnight and then sub-cultured in 50 ml tryptone media (0.015 g/l; Sigma) till mid-log phase ($OD_{600}$ at 200 rpm, 30°C).

### Construction of gene knockout strains

The $\Delta cheY$ mutant was gifted by Professor JS Parkinson and JPA804 strain by Professor Judith Armitage. Other knockout mutant strains were developed by using standard site-specific recombination method for inactivation of genes (*Datsenko and Wanner, 2000*; *Baba et al., 2006*). The desired gene was deleted by replacing it with kanamycin marker. The kanamycin marker was further removed by site-specific recombination around FRT region leaving behind scar of 80 base pairs, using a plasmid expressing an FLP recombinase. The desired deletion was confirmed by colony PCR with oligonucleotides as mentioned in *Table 2*. The details of the PCR with verification primers are given in *Figure 3—figure supplement 7*.

### Tethered cell experiment

Cells were centrifuged at 4000 rpm and washed twice with motility buffer (MB). MB is composed of $K_2HPO_4$, 11.2 g; $KH_2PO_4$, 4.8 g; $(NH_4)_2SO_4$, 2 g; $MgSO_4 \cdot H_2O$, 0.25 g; polyvinylpyrrolidone, 1 g; and EDTA, 0.029 g/l of distilled water. Cells were then resuspended in 1 ml MB and passed 75 times through a syringe fitted with 21-gauge needle to enable shearing of flagella. Cells were centrifuged at 4000 rpm for 10 min to obtain cells having short flagellar stub. The cell pellet so obtained was again diluted in MB.

Rectangular chambers were created with a microscope coverslip as base and flexible thin sheet of polydimethylsiloxane (PDMS) (Sylgard 184; Dow Corning) as the top wall separated by two parallel silicon rubber spacers of approximate thickness 400 µm. The thin sheets of PDMS were prepared by spreading the uncured polymer on polystyrene petri plates and then curing them for 8 hr at 50°C. The microscopic coverslip was cleaned with ethanol and smudged with thin layer of silicone oil (*Manson et al., 1980*). Use of silicone oil prior to antibody treatment in our case resulted in a more

**Table 1.** *Escherichia coli* strains used in this study.

| Strain | Relevant genotype | Parent strain | Comments, references |
|---|---|---|---|
| RP437 | | | Wild type, *Parkinson, 1978* |
| RP5232 | $\Delta cheY$ | RP437 | *Parkinson, 1978* |
| JPA804 | GFP MotB FliC-sticky | RP437 | *Leake et al., 2006* |
| MTKV01 | $\Delta trg$::Frt | RP437 | This study |
| MTKV04 | $\Delta ptsI$::Frt | RP437 | This study |

**Table 2.** Oligonucleotides used as primers for gene knockout.

| Designation | 5'– 3' Sequence | Function |
|---|---|---|
| PtsI- F1 | TAATTTCCCGGGGTTCTTTTAAAAATCAGTCACAAGTA AGGTAGGGTTATG ATTCCGGGGATCCGTCGACC | *ptsI* deletion |
| PtsI-R2 | AAGCAGTAAATTGGGCCGCATCTCGTGGATT AGCAGATTGTTTTTTCTTC TGTAGGCTGGAGCTGCTTCG | |
| Trg-F1 | GCCGATGACTTTCTATCAGGAGTAAACCTGGACGA GAGACAACGGTAATGATTCCGGGGATCCGTCGACC | *trg* deletion |
| Trg-R2 | GGGATCTGTCGATCCCTCCTTGAACATTTTCACACC GTAGCGAAACTAACTGTAGGCTGGAGCTGCTTCG | |
| PtsI- VF1 | CTGCTGCCCAGTTTGTAAAA | Confirming *ptsI* deletion |
| PtsI- VR2 | TTTACCAATGGTGCCGTCTA | |
| Trg-VF1 | AGGCATCCTATGAGGTTTCCT | Confirming *trg* deletion |
| Trg-VR2 | CTATCTCGTCAACTTACGGTTGAAT | |

stable interaction between anti-flagellin antibody and the glass slide. This ensured that tethered cells were not washed away when the ligands were introduced.

The aforementioned arrangement results in a chamber that is permeable to oxygen. The working volume of the chamber was approximately 80–100 µl. The chamber was first incubated for 30 min with a solution containing anti-flagellin antibody (Abcam) specific for bacterial flagellin. The solution was prepared by diluting the antibody 1000 folds in 50 mM Tris-HCl (pH 7.6). The chamber was washed gently with MB after which sheared cells were introduced into the chamber. An incubation time of 15–20 min was sufficient for the flagellum to tether to the antibody-coated coverslip surface. Unattached cells were washed and the channel was filled again with MB. Once the chamber was filled with MB, concentrated ligand solution (5000 µM glucose or 2Dg) of a known volume was then introduced gently through four holes punched in the PDMS sheet so that the average concentration of the final solution in the chamber is 1000 µM. The four holes surrounded the visualization region and were about 3–4 mm from the region. Experiments with color dyes showed that a combination of convective flow induced by the incoming dye solution and accompanying diffusion led to the dye reaching the visualization region within 30–45 s of introducing the dye. In order to measure the response of single flagellar motor to ligands, we first measured the response in MB and again after adding glucose/2Dg. Such pair-wise assessment removed any variability arising due to cell size and tethering geometry.

Experiments with ligands were performed at an ambient temperature of 25 ± 1°C. The imaging was done with an inverted microscope (Olympus IX71) fitted with 100× (1.4 NA), oil immersion objective. Area was scanned for rotating cells and videos were captured at an acquisition rate of 60 fps for 30–44 s using a CMOS camera (Hamamatsu Inc).

*Figure 2—figure supplement 1A* shows a schematic of the top and side view of a cell tethered to a glass coverslip (*Silverman and Simon, 1974*). *Figure 2—figure supplement 1B* presents an image of the cell along with two circles drawn by the Particle Tracker plugin of ImageJ software(NIH) for identifying the cell ends (*Sbalzarini and Koumoutsakos, 2005*). Particle detection and linking parameters were adjusted in such a way that only two circles fit the two ends of the rod-shaped bacteria. The coordinates of the centers of the two circles gave the orientation angle of the cell (*Figure 2—figure supplement 1A*). The change in orientation angle was determined for each subsequent frame (d$\theta$) and was used to determine the direction of rotation and the instantaneous rotation rate (Hz). Rotation rate values between −0.05 and 0.05 Hz were considered as a pause (*Eisenbach et al., 1990*) and were not used to determine the average rotational speed. Moving average of 10 instantaneous speeds were calculated so as to obtain the mean rotational speed and CCW bias. Since a cell undergoes several revolutions during the course of the experiment, the mean time required to complete a specified number of revolutions and the corresponding variance was determined from image analysis. The rotation of the cell was recorded and the angular position, obtained as a function of time, was used to determine the rotation speed and the rotation bias.

## GFP-MotB intensity measurement

For GFP-MotB intensity measurement, cells were prepared in the same manner as that for the tethered cell experiment except that the anti-flagellin antibody was not used in the flow chamber. This is because the mutant strain JPA804 (RP437 MotB-GFP-tagged cell) has sticky flagella (*Leake et al., 2006*) enabling them to easily bind to the substrate. Cells were imaged via spinning disk confocal microscopy (Perkin Elmer UltraVIEW system with Olympus IX71) under 100× (NA 1.4) oil immersion objective. First, bright-field videos were captured using a CMOS camera (Hamamatsu Inc) at an acquisition speed of 34 fps for 5 s for measuring rotation speed of cells. Immediately after this, the cells were excited with a 488 nm laser with exposure time of 200 ms (4–5 mW). Images of 16-bit resolution were captured at an acquisition rate of 6 fps for 1 s at different time points for rotating cells using an EMCCD camera (Hamamatsu Inc). Videos were analyzed using ImageJ software, and the fluorescence intensity of the stator proteins and motor speed were determined for the rotating cells. All the experiments were performed at 25°C. To eliminate the effect of varying background intensity from one experiment to another, the average pixel value obtained from three distinct locations (highlighted by the circles in *Figure 2—figure supplement 2*) around the cell was subtracted from the image. This ensured that the fluorescence intensity of the stator units was not influenced by the intensity variation in the background. This process was applied to all cells. The same cell was also visualized in brightfield to simultaneously capture the motor rotation speed (*Figure 2—figure supplement 2*). We observed that in some cases the cells did not rotate for the entire duration of the experiment. Such cases were not included. For analysis, we chose cells that were tethered at their polar ends and ignored cells that were tethered at the center of the body so that the motor torques at a fixed rotation rate are similar.

## Determination of swimming speed

Cells were centrifuged at 4000 rpm and washed twice with MB. Cells were added to each vial of ligand (1000 μM of glucose and 2Dg) and MB such that the cell density was low, about $10^6$–$10^7$ cells/ml. The cells were introduced into glass, rectangular microchannels (5 cm [L] × 1000 μm [W] × 100 μm [H]) via capillary action (*Deepika et al., 2015*) at different time points (*t*=0–15 min) for visualization and measurements. Both ends of microchannel were sealed with wax after introduction of cells. Trajectories of swimming bacteria were recorded with 40× (0.75 NA) objectives using darkfield microscopy. At each time point, six videos of about 15 s duration were recorded over a span of 3 min at a frame rate of 21 fps. Images were taken in the central region of the microchannel away from the channel walls. The trajectories of the cells were obtained using a commercial software, ImageProPlus. The data was analyzed using an in-house code written in MATLAB to obtain the swimming speed and cell orientation from more than 2500 cells for each condition. We considered only those cells that were within 1 μm of the focal plane with tracks longer than 0.5 s, which ensured that all out-of-plane motions were ignored by the analysis. A tumble event was identified when the swimming speed of the cell was below half the mean swimming speed and the change in the turn angle was greater than 4° between successive frames (at 21 fps) (*Alon et al., 1998*). The measured average swimming speed of 18.2 ± 7.9 μm/s (average ± standard deviation) and an average turn angle of 71° for RP437 cells dispersed uniformly in a microchannel containing plain MB are close to those observed for the same strain reported earlier, 18.8 ± 8.2 μm/s and 69°, by *Saragosti et al., 2011*. The details of the analysis have been described in earlier published reports (*Deepika et al., 2015*; *Karmakar et al., 2016*).

## Measurement of membrane potential

Cells preparation method was similar to that described before except that sodium phosphate buffer (NaMB) was used instead of the standard MB. NaMB contained $Na_2HPO_4$, 1.64 g; $NaH_2PO_4$, 0.47 g; NaCl, 8.77 g; and EDTA, 0.029 g/l of distilled water. The chemicals $Na_2HPO_4$, $NaH_2PO_4$, and NaCl were obtained from Merck while EDTA and CCCP were purchased from Sigma Aldrich. The cell pellet was dissolved in 500 μl of NaMB and 3.75 μl of the resulting solution was further diluted in NaMB containing ligand at a 1000 μM concentration such that a final cell density of $3.75 \times 10^8$ cells/ml was achieved.

 3,3-Diethyloxacarbocyanine iodide (($DiOC_2(3)$) from ThermoFisher Scientific) is a lipophilic cationic dye and was used as a membrane potential probe. $DiOC_2(3)$ was added at a final concentration of

30 μM to each sample vial after which samples were taken for fluorescence readings in microplate reader (Varioskan LUX plate reader) with excitation at 488 nm. Further, addition of EDTA facilitates dye uptake in gram-negative cells. Hence, 100 μM EDTA was introduced in NaMB.

DiOC$_2$(3) enters the cell as a monomer and binds to the membrane. Upon excitation by 488 nm wavelength light, the dye emits both green (520 nm) and red (620 nm) fluorescence. The green fluorescence intensity is dependent on the size of the cell while the red fluorescence intensity is dependent on both the membrane potential and the size of the cell. The ratio of red and green fluorescence (R/G) eliminates the dependence on cell size and directly correlates with the membrane potential of the cells. Membrane potential is negative on the inside of the membrane and so the magnitude of the negative charge correlates with the magnitude of R/G (*Novo et al., 2000*; *Kennedy et al., 2011*; *Alakomi et al., 2006*). CCCP is an ionophore which increases membrane permeability to protons and thereby eliminates membrane potential. In such a case, the R/G is the lowest and is used as a control in our experiments.

### pH homeostatsis *E. coli*

WT RP437 cells transformed with plasmid pMS201 having a gene with kanamycin resistance and green fluorescence protein (GFPmut2) were gifted by Professor Supreet Saini. The strain was used for determining pH homeostasis. Experiments were performed in the presence and absence of 20 mM sodium benzoate (weak permeant acid), which is known to collapse the pH difference across the cell membrane and equilibrate it to the pH value of the external medium. The cells were suspended in MBs with different pH ranging from 5.5 to 8.5. Samples were excited at 485 nm and emission was collected from 495 nm onward using a fluorescence spectrophotometer (Varioskan LUX plate reader). Peak emission was observed at 508 nm and the same was used for analysis.

## Acknowledgements

We thank Professor Supreet Saini for useful inputs on an earlier draft. Financial support from the Department of Science and Technology, India (SB/S3/CE/089/2013), and Department of Biotechnology, India (BT/PR7712/BRB/10/1229/2013), is acknowledged.

## Additional information

### Funding

| Funder | Grant reference number | Author |
|---|---|---|
| Department of Biotechnology | BT/PR7712/BRB/10/1229/2013 | Mahesh S Tirumkudulu |
| Department of Science and Technology | SB/S3/CE/089/2013 | Mahesh S Tirumkudulu |

The funders had no role in study design, data collection and interpretation, or the decision to submit the work for publication.

### Author contributions

Farha Naaz, Megha Agrawal, Conceptualization, Data curation, Software, Formal analysis, Validation, Investigation, Visualization, Methodology, Writing - original draft, Writing - review and editing; Soumyadeep Chakraborty, Formal analysis, Validation, Investigation, Visualization, Methodology; Mahesh S Tirumkudulu, Conceptualization, Resources, Data curation, Software, Formal analysis, Supervision, Funding acquisition, Methodology, Writing - original draft, Project administration, Writing - review and editing; KV Venkatesh, Conceptualization, Resources, Supervision, Funding acquisition, Writing - original draft, Writing - review and editing

### Author ORCIDs

Mahesh S Tirumkudulu https://orcid.org/0000-0002-7046-8069

## Decision letter and Author response
Decision letter https://doi.org/10.7554/eLife.62848.sa1
Author response https://doi.org/10.7554/eLife.62848.sa2

## Additional files
### Supplementary files
• Transparent reporting form

### Data availability
All data generated or analysed during this study are included in the manuscript and supporting files. Source data files have been provided for Figures 2, 3 and 4.

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

## Appendix 1

### Supplemental information

*Figure 3—figure supplement 1* presents the rotation rate and the CCW bias for WT cells before and after they were exposed to glucose. The measurements were made 5 min after exposure to glucose. As expected, both the rotation rate and CCW bias are higher compared to the pre-stimulus value. Note that rotation rate for glucose at 15 min (*Figure 3—figure supplement 3B*) is higher than that at 5 min. The corresponding measurements for 2Dg have been reported in *Figure 3—figure supplement 3C* and *Figure 3—figure supplement 4C*.

*Figure 3—figure supplement 2* presents the normalized rotation speed and the CCW bias for five cells after exposure to 1000 μM of glucose and 2Dg. As expected, the rotation speed and the CCW bias increase with time. While the CCW bias in case of 2Dg returned to its pre-stimulus value, the same was not observed for glucose even after 15 min exposure. The variation of the measured CCW values among cells is presented in *Figure 3—figure supplement 2B,D*.

*Figure 3—figure supplement 3* shows the distribution of rotational frequency in CCW direction for at least 20 cells each of WT, $\Delta trg$, and $\Delta ptsI$ cells measured in MB, and 1000 $\mu$M of glucose and 2Dg. The mean rotational speed of WT in MB was measured to be about 4.14 $\pm$ 1.65 Hz (*Figure 3—figure supplement 3A*), which compares well with the previous studies (*Eisenbach et al., 1990*) . Experiments with glucose resulted in a rotation speed of 5.95 $\pm$ 1.69 Hz at 15 min (*Figure 3—figure supplement 3B*), an increase of 44% in motor speed compared to that observed in MB. At the intermediate time of 5 min, motor speed of 4.82 $\pm$ 1.74 Hz (*Figure 3—figure supplement 1B*) was obtained which corresponds to about 25% increase as compared to MB, which is inline with the trend observed for run speeds. In comparison, an increase of 27% (5.28 $\pm$ 1.74 Hz) was observed at 5 min in the presence of 2Dg (*Figure 3—figure supplement 3C*). At longer times, the rotation speed decreased significantly, as also observed in case of run speed, due to the lack of energy source. These results clearly demonstrate that sensing alone can enhance the motor speed leading to an increase in run speed. These measurements also yield CCW bias for the three cases. A CCW bias of 0.90 $\pm$ 0.09 was observed for glucose at 15 min (*Figure 3—figure supplement 4B*) and 0.78 $\pm$ 0.16 for 2Dg at 5 min (*Figure 3—figure supplement 4C*), compared to 0.68 $\pm$ 0.16 in MB (*Figure 3—figure supplement 4A*).

Tethered cell experiments were also performed with $\Delta trg$ mutant strain. *Figure 3—figure supplement 3D–F* represents the distribution of rotation speed in CCW direction in MB, glucose, and 2Dg, respectively. It can be clearly seen that the mean rotation speed increased by 30% (3.45 $\pm$ 1.80 Hz) in glucose as compared to MB, whereas no significant variation was observed when the cells were exposed to 2Dg. This reiterates the role of Trg receptor in modulating the motor speed. The observed increase in rotation speed in the presence of glucose is caused by the glucose PTS uptake mechanism. The CCW bias mimicked the trend observed for the rotation speed, wherein CCW bias increased by 13% in glucose related to MB, with no change in 2Dg (*Figure 3—figure supplement 4D–F*).

A similar exercise performed with $\Delta ptsI$ mutant strain (*Figure 3—figure supplement 3G–I*) showed a 24% increase in rotation speed in the presence of glucose compared to MB. However, the small increase (16%) observed in 2Dg was not statistically significant (p>0.05). Further, the change observed in the $\Delta ptsI$ strain compared to $\Delta trg$ strain was lower. A similar trend was observed for the CCW bias (*Figure 3—figure supplement 4G–I*).

*Figure 3—figure supplement 4* presents the CCW bias for the WT, $\Delta trg$, and $\Delta ptsI$ cells measured in MB, and 1000 $\mu$M of glucose and 2Dg. In case of glucose, the CCW bias increases with time and remains higher than the pre-stimulus value for all strains even after 15 min. In case of 2Dg, while the bias returns to the pre-stimulus value within 5 min for both $\Delta trg$ and $\Delta pstI$, it takes longer (between 5 and 10 min) in case of WT (*Figure 3—figure supplement 2*).

*Figure 3—figure supplement 5* presents the variance of the rotation rate for increasing number of revolutions recorded for a single WT cell exposed to 1000 μM of glucose and 2Dg. At a fixed time, the variance increases with number of revolutions but the magnitude of the slope decreases with time in the presence of glucose. However, in case of 2Dg, the slope decreases initially and then increases after 5 min. Decreasing variance indicates increasing number of torque-generating units

leading to a smoother rotation of the motor. These results are consistent with the behavior observed for both single motor and a population.

*Figure 3—figure supplement 6* presents the fluorescent intensity corresponding to the cytoplasmic pH. In the absence of sodium benzoate, the intensity value is independent of the external pH indicating that the cell maintains a constant pH irrespective of the external conditions. In the presence of sodium benzoate, which causes the pH difference to collapse, the fluorescence intensity varies with change in external pH. These observations are in agreement with previously published results (*Wilks and Slonczewski, 2007*; *Hansen and Atlung, 2011*; *Slonczewski et al., 1981*). As a consequence, all changes in the PMF observed in our experiments, which were all performed near neutral pH, are due to changes in membrane potential.

