## [Decision Letter]

**Acceptance summary:**

This paper will be of interest to scientists interested in chemotaxis and bacterial motility. Using *E. coli*, it demonstrates, through a range of experimental methods, that the widely held view that the signals associated with ligand binding in chemotaxis serve to bias the direction of rotation of the motor without changing its speed is not correct, and that there is a demonstrable transient change in speed.

**Decision letter after peer review:**

Thank you for submitting your article "Ligand Sensing Enhances Bacterial Flagellar Motor Output via Stator Recruitment" for consideration by *eLife*. Your article has been reviewed by 3 peer reviewers, one of whom is a member of our Board of Reviewing Editors, and the evaluation has been overseen by Suzanne Pfeffer as the Senior Editor. The following individual involved in review of your submission has agreed to reveal their identity: Richard M Berry (Reviewer #3).

The reviewers have discussed the reviews with one another and the Reviewing Editor has drafted this decision to help you prepare a revised submission.

We would like to draw your attention to changes in our policy on revisions we have made in response to COVID-19 (https://elifesciences.org/articles/57162). Specifically, when editors judge that a submitted work as a whole belongs in *eLife* but that some conclusions require a modest amount of additional new data, as they do with your paper, we are asking that the manuscript be revised to either limit claims to those supported by data in hand, or to explicitly state that the relevant conclusions require additional supporting data.

Summary:

This is an interesting study reporting an increase in the rotation speed of the *E. coli* flagellar motor upon the sensing of a non-metabolizable glucose analog (2Dg) by the cell. The authors conclude that this increase is due to an increase in the number of torque-generating stator complexes that drive the motor. Knockout of the trg gene abolished this effect, suggesting that sensing of 2Dg by the Trg chemosensor is responsible. Involvement of membrane potential, the PTS pathway, and the chemotaxis response regulator CheY is ruled out. The manuscript is well-written, and the data are convincing. But the mechanism remains unclear.

Essential revisions:

1. While reported as significant at P = 0.01 level, the GFP-MotB data would be much improved by more repetitions, which should not be difficult to achieve. Normally at least 20-30 motors should be included, at least in the main experiments.

2. Apparent motor and swimming speed increases during chemotactic responses have been attributed in the past to un-resolved switching and partial switching events. Typically, care should be taken to rule out these possibilities and to demonstrate that this has been done. At minimum, speed recordings should be shown that demonstrate the time resolution for detecting short switching events, and characterizing the distributions of event lengths. Then analysis to test whether reported average speed changes could be attributed to missed events. However, the control experiment with CheY deletion makes these steps unnecessary, as it demonstrates that the observed speed changes are independent of switching. This should be made clear and the importance of the CheY experiment emphasized in the text.

3. The authors hint towards the involvement of c-di-GMP signaling via the YcgR protein. This hypothesis can be tested by knocking down the ycgr gene and repeating the assay, but this has not been done or reported. Addition of these data to the manuscript would make the paper significantly stronger.

4. Do other chemoreceptors (Tar, Tsr, Tap) also act in the same way with their respective ligands? It would be useful to know if this effect is specific to Trg or if it is also found in the other chemoreceptors.

5. In figure 3C, what is the reason that the GFP intensity and the speed do not have the same range? In other words, why is the slope not equal to 1? Since there is 1:1 correspondence between the number of MotB and the number of GFP, shouldn't the slope be 1?

6. The authors do not cite or discuss the recent literature on load-dependent stator remodeling (e.g. PMIDs: 29183968, 31142644). It would be helpful to have a more in-depth discussion on how the observed stator unit recruitment relates to stator remodeling in response to load.

7. The authors should make quantitative the changes in chemotactic behaviour that would be expected as a consequence of the motor speed changes revealed in this research. That is, can the authors put some numbers into a standard analysis of run-and-tumble dynamics to quantify any improvement in chemotactic efficiency or speed under such changes?

---

## [Author Response]

Essential revisions:1. While reported as significant at P = 0.01 level, the GFP-MotB data would be much improved by more repetitions, which should not be difficult to achieve. Normally at least 20-30 motors should be included, at least in the main experiments.

We have now performed experiments so as to have more than 20 motors for each case (MB: 21 cells, Glu: 21 cells, 2Dg: 22 cells). We have updated Figure 3.

2. Apparent motor and swimming speed increases during chemotactic responses have been attributed in the past to un-resolved switching and partial switching events. Typically, care should be taken to rule out these possibilities and to demonstrate that this has been done. At minimum, speed recordings should be shown that demonstrate the time resolution for detecting short switching events, and characterizing the distributions of event lengths. Then analysis to test whether reported average speed changes could be attributed to missed events. However, the control experiment with CheY deletion makes these steps unnecessary, as it demonstrates that the observed speed changes are independent of switching. This should be made clear and the importance of the CheY experiment emphasized in the text.

Yes, we agree that observed motor speeds changes are not affected by motor reversals as shown by the experiments with the CheY deletion mutant strain. We have emphasized this point on lines 199-201 (page 9):

“The similarity of the observed trend with WT indicates that CheY-P has no role in the observed modulation in motor speed and the observed speed increases are independent of switching events.”

Note that we again emphasize the same in the Discussion section, page 10, line 257 onwards.

3. The authors hint towards the involvement of c-di-GMP signaling via the YcgR protein. This hypothesis can be tested by knocking down the ycgr gene and repeating the assay, but this has not been done or reported. Addition of these data to the manuscript would make the paper significantly stronger.

Our preliminary swimming speed study with YcgR gene knockout mutant are promising. However, detailed tethered cell studies with GFP-MotB tagged WT having YcgR gene deletion will take time to perform and analyze. The results of such a study would be reported in a separate publication. Our focus in this manuscript is to show that the motor speed modulation can occur due to sensing of ligand and is caused by stator recruitment.

4. Do other chemoreceptors (Tar, Tsr, Tap) also act in the same way with their respective ligands? It would be useful to know if this effect is specific to Trg or if it is also found in the other chemoreceptors.

We have initiated experiments on the Tar receptor which senses α methyl aspartate. The preliminary results do show that the swimming and motor speeds increase in the response to the ligand. Again, we need to perform many more experiments and that is expected to take time. The results will be reported in a separate publication.

5. In figure 3C, what is the reason that the GFP intensity and the speed do not have the same range? In other words, why is the slope not equal to 1? Since there is 1:1 correspondence between the number of MotB and the number of GFP, shouldn't the slope be 1?It is required that the GFP intensity (measured in arbitrary units) be directly proportional to the speed but the slope need not be equal to one. Since the values are normalized with the pre-stimulus values, the value of the slope would also be effected with the intensity and speed values used for normalization. For example, if we assume that the speed *V* is directly proportional to intensity, *I*, we have V=kI. When the intensity doubles so will the velocity but the ratio, V/I, will always be k.*6. The authors do not cite or discuss the recent literature on load-dependent stator remodeling (e.g. PMIDs: 29183968, 31142644). It would be helpful to have a more in-depth discussion on how the observed stator unit recruitment relates to stator remodeling in response to load.*

We have added these two references in the paper and discussed them on page 2, lines 56-59:

“Consequently, the bacteria is able to vary the number of motor-bound stators depending on the external mechanical load [Tipping et al., 2013a; Lele et al., 2013; Nord et al., 2017; Wadhwa et al., 2019] although not all binding sites are necessarily occupied even at high load conditions due to the transient nature of stator binding.”

7. The authors should make quantitative the changes in chemotactic behaviour that would be expected as a consequence of the motor speed changes revealed in this research. That is, can the authors put some numbers into a standard analysis of run-and-tumble dynamics to quantify any improvement in chemotactic efficiency or speed under such changes?

The well-known RTBL model of Rivero et al. (Chem Engg Sci, 1989), which is based on the Keller-Segel model, links the population-level chemotactic response to the individual microscopic variables of cell’s swimming speed, persistence time and temporal receptor occupation. The drift velocity (also known as the chemotactic velocity) of the population is directly proportional to the product of the swimming speed and a function that depends on the tumbling frequency, spatial gradient of chemoattractant and occupancy of chemoreceptor [Equation 22 of Rivero et al. (1989)]. For small gradients, the expression simplifies such that the drift velocity varies as the square of the swimming speed [Equation 25 of Rivero et al. (1989)]. These results would suggest that even a 10% increase in the swimming speed could cause the drift velocity to increase anywhere between 10-20%, which is a significant improvement in the chemotactic efficiency.